# An Overview of Medium Access Control and Radio Duty Cycling Protocols for Internet of Things

**Farhan Amin** [1] , **Rashid Abbasi** [2,*] , **Salabat Khan** [3] and **Muhammad Ali Abid** [4]

1 Department of Information and Communication Engineering, Yeungnam University, Gyeongsan 38541, Republic of Korea
2 School of Electrical Engineering, Anhui Polytechnic University, Jiujiang District, Wuhu 241000, China
3 The College of Computer Science and Software Engineering, Shenzhen University, Shenzhen 518060, China
4 Faculty of Smart Engineering, the University of Agriculture, Dera Ismail Khan 29220, Pakistan
* Correspondence: rashidd.abbasi@gmail.com

**Abstract:** The Internet of Things (IoT) applications such as smart grids, smart agriculture, smart cities, and e-healthcare are popular nowadays. Generally, IoT end devices are extremely sensitive to the utilization of energy. The medium access control (MAC) layer is responsible for coordination and access of the IoT devices. It is essential to design an efficient MAC protocol for achieving high throughput in IoT. Duty cycling is a fundamental process in wireless networks and also an energy-saving necessity if nodes are required to operate for more than a few days. Numerous MAC protocols along with different objectives have been proposed for the IoT. However, to the best of our knowledge, only limited work has been performed dedicated to covering MAC and radio duty cycling (RDC). Therefore, in this study, we propose a systematic cataloging system and use if to organize the most important MAC and RDC proposals. In this catalog, each protocol has been categorized into main ideas, advantages, applications, limitations, innovative features, and potential future improvements. Our critical analysis is different from previous research studies, as we have fully covered all recent studies in this domain. We discuss challenges and future research directions.

**Keywords:** internet of things; radio duty cycling; short-range protocols; long-range protocols; MAC layer protocols; TinyOS



## 1. Introduction

In the Internet of things (IoT), physical objects such as actuators and sensors are connected to the Internet [1]. The smart home, smart factory, and smart healthcare are key examples using this approach. All these applications help in improving the quality of human life. Figure 1 illustrates the conceptual view of the IoT [1]. In this figure, the IoT end devices are connected to physical objects. These devices send and receive the data to the IoT via a cloud server. This control and connectivity are ensured using communication technologies. These technologies enable connectivity between different machines and also different applications. These are referred to as IoT gateways. Key examples include Wi-Fi IEEE 802.11 [2], Bluetooth IEEE 802.15.1 [3], IEEE 802.15.4/802.15.4e ZigBee [4], and IEEE 802.11p WAVE sub 1 GHZ [5]. Generally, IoT end devices are extremely sensitive to the utilization of energy. These applications mostly rely on the network system's lifetime. The Medium access control (MAC) protocol ensures the successful transmission of data [6]. The MAC protocol is characterized as follows [7].

(1) Throughput: The Mac protocols ensure high throughput. Different IoT applications require different throughputs such as temperature and fire detection.
(2) Scalability: Network scalability is considered with the addition and removal of IoT end devices.
(3) Latency: The MAC protocol also helps in reducing the latency.

(4)   Energy Management: The IoT end devices are highly energy-constrained. Therefore, the MAC protocol ensures a highly energy-efficient solution for IoT devices.

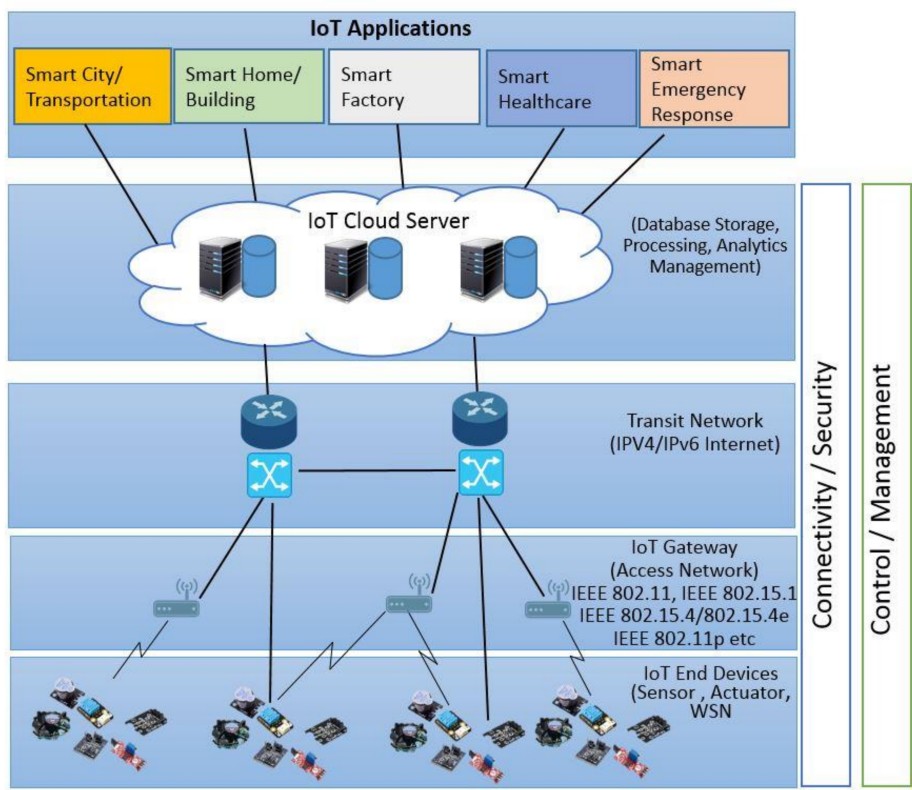

**Figure 1.** The conceptual view of the Internet of things [1].

Generally, the MAC protocols use Radio duty cycling (RDC). The RDC combines different techniques and communication protocols. This ensures the efficient utilization of energy in wireless sensor networks. There are a variety of MAC and RDC protocols available for the WSN. The researchers are working to get insights from these protocols for different operating systems (OS). Currently, the most common OS are TinyOS [8] and OpenWSN [9]. Many protocols and Os are developed to fulfill the requirements of industrial and healthcare applications [10]. These protocols support a protocol stack, an open application program interface (API). This provides support for various programming languages such as C/C++, Javascript and Python. [11]. For instance; the component-based architecture enables rapid innovation and implementation in TinyOS. In addition, it minimizes the code size. Its component library includes network protocols, sensor drivers, distributed services, and data acquisition tools. Another unique feature of TinyOS is the power of event-driven execution. It enables fine-grained power management. It allows the scheduling and flexibility of the unpredictable nature of wireless communication. OpenWSN provides an open-source implementation. This implementation helps academia and industry to verify the applicability of different standards in IoT. In addition, it enables highly reliable mesh networks [11]. Table 1 demonstrates the comparison and the key features of both OS. These features are based on support, no support, or partial support. In this table, we examine which OS has support for multithreading and modularity or not, etc.; for instance, OPENWSN has support. These Os are widely used nowadays due to their small size and high power. The MAC layer is responsible for the coordination and also to enable access among the IoT devices. The power control mechanism is also ensured by using the MAC protocol. Therefore, it is mandatory to design an efficient medium access control (MAC) protocol for achieving high throughput, ensuring, efficient bandwidth allocation, and the nodes should be synchronized with time. A very limited range of re-

search proposals have been reported in this area. Therefore, in this study, we propose a systematic catalog and organize the most important MAC and RDC proposals along with key responsibilities. Table 2 offers a comparison of the most recent research and our study. For instance, the most recent study conducted by Shreya et al. in [6] in partially covers contention-based protocols. They did not discuss the hybrid and contention-based protocols. Similarly, Balobaid et al. [7] cover only Single-channel protocols but they did not discuss the hybrid MAC protocols in detail. Olympia et al. in [12] explored hybrid and partial signals channel protocols. Kakria et al. in [13] only cover single channel protocols. Finally, Arain et al. in [14] only cover multichannel contention-free protocols. In Table 2 a summary and a comparison of our study and recent surveys are shown. We can easily see that many researchers have not or briefly covered contention-based, contention-free, and hybrid-based protocols. In contrast, we have covered all. We present the following contributions to the research community. In this study, we highlight the importance of the higher layer and the RDC in perspectives for the IoT.

**Table 1.** Tinyos and Openwsn.

| OS | TinyOS | OpenWsN |
|---|---|---|
| Min RAM | <1 kB | <10 kB |
| Min ROM | <4 kB | <8 kB |
| C/C++ support | - | ✓ |
| Multithreading | ● | ✓ |
| MCU w/oMMU | ✓ | ✓ |
| Modularity | - | ✓ |
| Real Time | - | ✓ |

✓Full support; ● Partial support; - No-support

**Table 2.** Summary and comparison of our study and recent surveys.

| Approaches | Contention-Based | | | | | | Contention-Free | | | | Hybrid MAC | |
| | Sender-Initiated-Protocols | | | | | | Scheduled-Based Protocols | | | | | |
| | Single-Channel Protocols | | Multi-Channel Protocols | | | | Centralized Protocols | | Distributed Protocols | | Hybrid MAC | |
| | | | Static Channels | | Dynamic Channels | | | | | | | |
| | T | O | T | O | T | O | T | O | T | O | T | O |
|---|---|---|---|---|---|---|---|---|---|---|---|---|
| Our Study | ✓ | ⊠ | ✓ | ⊠ | ✓ | ✓ | ✓ | ⊠ | ✓ | ✓ | ✓ | ⊠ |
| Shreya [6] | ⊠ | ⊠ | ⊠ | ⊠ | ⊠ | ⊠ | ⊠ | ⊠ | ⊠ | ⊠ | ⊠ | ⊠ |
| Balobaid [7] | ✓ | ⊠ | ⊠ | ⊠ | ⊠ | ⊠ | ⊠ | ⊠ | ⊠ | ⊠ | ✓ | ⊠ |
| Olempia et al. [12] | ✓ | ⊠ | ⊠ | ⊠ | ⊠ | ⊠ | ⊠ | ⊠ | ⊠ | ⊠ | ✓ | ⊠ |
| Kakria et al. [13] | ✓ | ⊠ | ⊠ | ⊠ | ⊠ | ⊠ | ⊠ | ⊠ | ✓ | ⊠ | ⊠ | ⊠ |
| Arain et al. [14] | ⊠ | ⊠ | ✓ | ⊠ | ✓ | ⊠ | ✓ | ⊠ | ✓ | ⊠ | ⊠ | ⊠ |
| Rajandekar et al. [15] | ⊠ | ⊠ | ⊠ | ⊠ | ⊠ | ⊠ | ✓ | ⊠ | ✓ | ⊠ | ✓ | ⊠ |
| A. Laya et al. [16] | ⊠ | ⊠ | ⊠ | ⊠ | ⊠ | ⊠ | ⊠ | ⊠ | ⊠ | ⊠ | ✓ | ⊠ |
| Alfayez et al. [17] | ✓ | ⊠ | ⊠ | ⊠ | ⊠ | ⊠ | ⊠ | ⊠ | ⊠ | ⊠ | ✓ | ⊠ |
| Ricardo et al. [18] | ✓ | ⊠ | ⊠ | ⊠ | ⊠ | ⊠ | ⊠ | ⊠ | ⊠ | ⊠ | ✓ | ⊠ |

T = TinyOS. O = OpenWSN.

We propose a classification catalog and presente the most recent state-of-the-art protocols. This catalog classifies the protocols into three classes: (1) contention-based protocols; (2) contention-free protocols; and (3) hybrid protocols, which incorporate the advantages of

contention-free and contention-based protocols while trying to alleviate their weaknesses. The effectiveness of these protocols in the context of IoT and the application scenarios is discussed, along with their advantages and weakness. We have classified the RDC protocols into two classes; for instance; Synchronous Low duty cycle protocols and Asynchronous Low duty cycle protocols. The critical problems and the challenges for the designing of IoT MAC/RDC protocols are discussed. Our critical analysis is different from previous research studies, since most studies have partially covered or not covered various aspects but we have fully covered all recent research in this area. Finally, some challenges, open issues, and recommendations for future work are provided, which may help to improve available schemes or the design of more innovative MAC/RDC protocols for IoT.

The organization of this paper is as follows. In Section 2, we discuss the most recent research proposals in this area. The most recent proposals have been classified in a systematic catalog and are explained in Section 3. In this section, each study is discussed along with its advantages and disadvantages. We have discussed RDC protocols in Section 4. Section 5 explains challenges and suggests future research directions. Section 6 presents the conclusions of our paper.

## 2. Related Work

In recent years, many researchers have focused on the design and development of MAC [17] and RDC [18] protocols for wireless networks. Most of the research has focused on contention-based, contention-free, and hybrid-based protocols. For instance, Balobaid in [7] provided a comparative study, discussing asynchronous, scheduling-based, contention-based, TDMA, and hybrid protocols. The study did not discuss standard protocols, such as Sensor MAC (S-MAC) [19], Berkeley MAC (B-MAC) [20], Time out MAC (T-MAC) [21], and Micro-MAC (μ-MAC) [22–24]. The study lacks a few of the recent protocols, such as demand wakeup MAC (DW-MAC) [25], Routing-Enhanced MAC (R-MAC) [26], Dynamic MAC (D-MAC) [27], Query MAC (Q-MAC) [28] and Scheduled channel polling MAC protocols (SCP-MAC) [29]. Similarly, Olympia et al. in [12] focused on the energy-efficient contention-based asynchronous and hybrid-based MAC protocols for WSNs. They performed a comparison of energy efficiency, throughput, latency, and fairness [12]. The contention-based protocols are suitable for small-scale applications, and on the other hand, the hybrid-based protocols are suitable for large-scale applications. The authors discussed energy-efficient contention-based MAC protocols, such as S-MAC, T-MAC, B-MAC, and hybrid-based MAC protocols, such as Advertisement MAC (ADV-MAC) [30], X-MAC [31], energy-efficient MAC (EE-MAC) [32]. However, this study lacks a discussion of some recent MAC protocols such as μ-MAC, SCP-MAC, and WISE MAC [33]. Kakra and Aseri in [13] discussed the evaluation of new contention-free synchronous protocols for WSN, such as SCP-MAC, DW-MAC, T-MAC, and D-MAC. They also discussed a few duty cycle MAC protocols, such as the S-MAC and Q-MAC protocols. They only focused on synchronous MAC protocols for WSN. Another limitation of this study is that only MAC protocols for TinyOS are discussed. Later, two studies [14,15] presented subject-wise categorization criteria for the grouping of various surveys. Arain and Ghani et al. [14] deliberated on subject-wise categorization of over 30 surveys on the MAC protocols for WSN. The authors provided an extensive survey of over two hundred MAC protocols for WSN. Generally, this survey is a comprehensive and fairly detailed work covering various MAC protocols for WSN, such as D-MAC, Q-MAC, Zebra-MAC (Z-MAC), Wise-MAC, and SCP-MAC. One of the limitations is that of the over 30 surveys, almost half of the MAC protocols did not discuss limitations. Additionally, none of discussed MAC protocols were based on OpenVPN. Similarly, Rajandekar et al. in [15], presented the issues related to efficient, scalable, and fair channel access for machine-to-machine (M2M) communication, reviewing current protocols. This paper provides ongoing standardization efforts and opens up issues for future research. The authors discuss various protocols for distributed point coordination Function-M (DPCF-M), Code Expanded Random access (CERA), Fast adaptive slotted ALOHA (FASA), and the performance evaluation of reservation frame-slotted

ALOHA (RFSA). Laya et al. in [16] anticipate an overview of recent MAC solutions for the IoT, describing current limitations and envisioned challenges for the near future. They identify a family of simple algorithms based on distributed queuing (DQ) [16], which can operate for an infinite number of devices generating any traffic load and pattern. The authors describe the first demo of DQ for IoT. Research on DQ applied in communication networks has already been carried out, showing how powerful this technology could also be for the IoT. The authors discussed modern technologies, e.g., Zigbee, BLE, RFID, and WIFI technologies. They emphasized contention-based channel access protocols for IoT, such as Pure ALOHA, slotted Aloha, unslotted CSMA/CA, and Slotted CSMA/CA. This work less distinctively reviews various algorithms based on DQ [34,35]. Alvarez et al. in [36] reviewed various RDC and MAC protocols under duty cycled asynchronous and synchronous protocols. They analyzed the strength and weaknesses of numerous MAC protocols, such as S-MAC, T-MAC, R-MAC, X-MAC and Wiseman. This research aimed to review the relevant MAC protocols to apply a duty cycle function, as the current MAC solutions are not solving the communication delay. The authors do not discuss sender-initiated and receiver-initiated asynchronous MAC and RDC protocols for LWSNS [37]. Ricardo et al. [38] presented a survey on duty cycle mechanisms. The authors reviewed synchronous, asynchronous, and semi-synchronous protocols. These protocols are primarily divided into rendezvous, skewed, preamble sampling, and schedule-based and elected cluster-head approaches. A comprehensive survey on the design challenges for MAC protocols can be found in [38]. Although this was good work since the authors focused not only on various techniques for energy conservation such as S-MAC, D-MAC, and μ-MAC, but also on some application-based scenarios of duty cycling. This survey article does not discuss scheduled-based and hybrid-based MAC protocols such as T-MAC, however. It is concluded from this section that most researchers have not focused on contention-based, contention-free, and hybrid-based protocols. Therefore, we propose a classification catalog to fill this gap. The proposed classification is explained in the next section.

## 3. Proposed Classification

We propose a systematic cataloging classification, as shown in Figure 2. In this classification, the MAC protocols are classified based on their channel access strategy, i.e., contention-based, contention-free, and hybrid-based methods. At the next level, the contention-based MAC protocols are further divided into sender-initiated and receiver-initiated protocols. Single-channel and multi-channel protocols are two separate branches of contention-free methods. At the next level, the multichannel protocols are further subdivided into static and dynamic channels. The complete details of our proposed classification are given below.

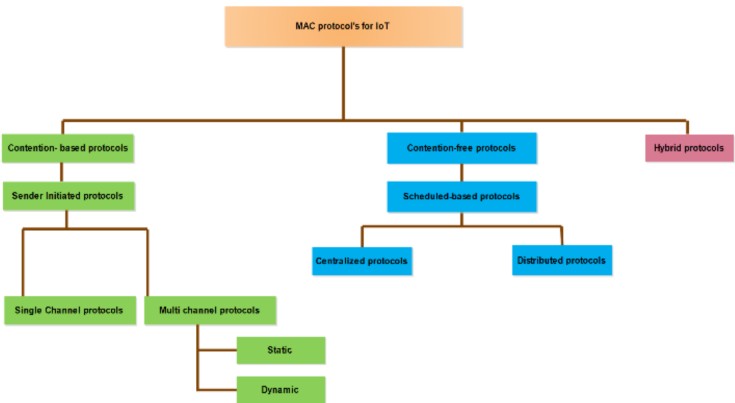

**Figure 2.** The proposed classification.

### 3.1. Contention-Based Protocols

In this method, the users contend with each other for channel access to transmit the data [17]. The user which wins the contention occupies the medium for some time, and

transfers the data. On the other hand, the other users stay silent, monitoring the channels to find a free channel. The flow chart explaining this mechanism is shown in Figure 3. These protocols generally use the CSMA/CA strategy for accessing the medium [17]. In CSMA/CA, a node tries to find a free channel before any transmission. If the channel is busy, it defers transferring the data [17]. These protocols are further divided into sender-initiated and receiver-initiated protocols. The details are given below.

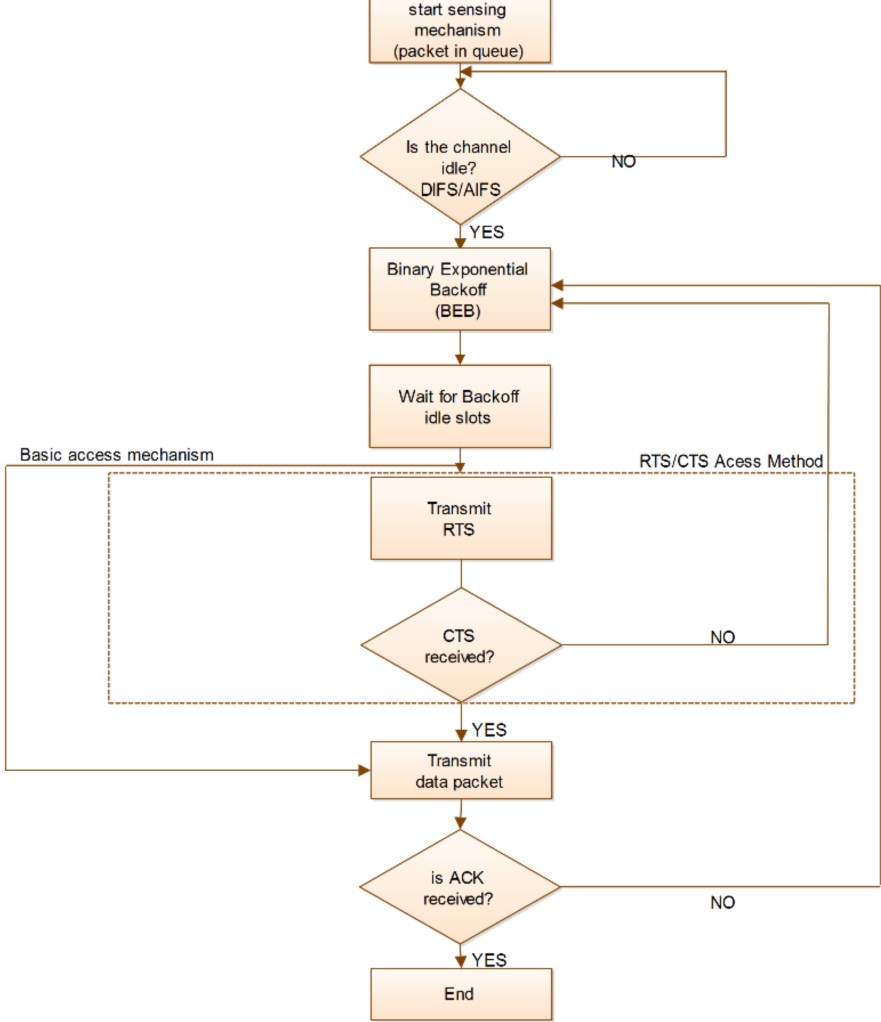

**Figure 3.** The flow chart of contention-based protocols.

### 3.2. Sender-Initiated Protocols

In sender-initiated protocols, the packet transmissions are initiated by the sender [17]. The sender-initiated protocols are further subdivided into two single-channel and multi-channel protocols.

### 3.2.1. Single Channel Protocols

In this method, only one channel is used to share the information, e.g., sending control messages [19]. Various researchers have devoted their hard work to this area. For instance; Wei Ye. et al. [19], introduced a protocol named Sensor-MAC (S-MAC). S-MAC is one of the best-known protocols designed for WSNs. It is designed for TinyOS. The key design goals of this protocol are to reduce energy consumption, support good scalability, and be self-configurable for WSN [19]; it works on the synchronization of sleep and awake schedules. In listening, a mode is known as the "awake schedule" and, in the "sleep schedule", the nodes are in sleep mode. This protocol also uses the CSMA/CA mechanism.

The working mechanism is as follows. It periodically sleeps and awakens and cycles to bring down idle listening [19]. The time is divided into small cycles and every period is framed by the listening period and the sleep period. The schedule of sleep and awake in S-MAC is illustrated in Figure 4. In Figure 4a, in sleep mode, a sensor node must turn off its radio. The objective of the turnoff radio is to restore energy. Generally, this energy is consumed during the wake-up [19]. The sensor node is going to sleep mode periodically for a certain interval of time. The sensor node wakes up after some time and listens to the available channel. This confirms whether any node is trying to send data or not. If a channel is free then data transmission is initiated [19]. Figure 4b demonstrates the general structure of the S-MAC. In this figure, a node is required to listen at regular time intervals to hear the channel. If a node is not transferring the data then a large amount of energy will have been lost. The duty cycle is helpful to reduce it. The duty cycling in S-MAC is performed by splitting each frame into awake and sleep intervals. Figure 4c explains this mechanism. The awake interval frame consists of 'SYNC and DATA'. Each node first listens to the channel. At the time of initialization, it is performed for a fixed duration of time. Then, a node calculates its sleep and awake timing. If no "SYNC" packet has been received at a particular interval, it broadcasts a "SYNC" packet to all other neighboring nodes. To achieve minimal energy utilization, the participating nodes continuously exchange RTS (Request to send) and CTS (Clear to send) packets. The drawback of the S-MAC is that it only controls the local interactions of the node in a network. The S-Mac protocol is useful for smart space, medical systems, and robotic exploration. This protocol is also used for smart grid applications. Similarly, Demirkol et al. [20] introduced Berkeley Media Access Control (B-MAC) for WSN. This scheme consists of sampling with a medium at fixed time intervals as shown in Figure 5a. The listening to a channel is performed for a period [20]. The node at first samples the medium. The objective is to check whether a node is willing to communicate or not [20]. If a node has a packet to send, the sender node senses the medium to see if it is free [20]. It takes a back off, and then sends a long wakeup preamble followed by a data packet, as shown in Figure 5b. When a receiver wakes up, it senses the medium, and if it detects any preamble, it turns on its radio and waits for the preamble to end. On completion, if the data packet is intended for a node, it receives the full data packet. Otherwise, it ignores the packet and goes to sleep. It is reconfigurable by networks and its implementation is simple and requires a small RAM size. [19]. The latency is increased due to the long-size preamble. This protocol is useful in environmental monitoring applications such as as temperature, etc. Timeout-MAC (T-MAC) in [21] is proposed to enhance the poor results of the S-MAC. The sensor node in T-MAC goes to the rest period if there is no traffic for a specific period [21]. The sensor node checks the channel for a certain time, if there is nothing in the channel, it goes to sleep mode. The sleep-wake-up schedule of the T-MAC is shown in Figure 6. The node listens and transfers when they are in an active period. T-MAC's significant flaw is an early resting issue in which the nodes may rest according to their time and data may get lost, particularly for long messages [22]. R-MAC, described in [26], uses multi-hop forwarding in a single operational cycle by shifting the data transmission to the sleep period. A cycle is divided into SYN, DATA, and sleep [29]. A single operational cycle is presented in Figure 7. The key application for this protocol for battery use in WSN. In this protocol, energy efficiency and multi-hop data forwarding are guaranteed.

### 3.2.2. Multi-Channel MAC Protocols

The frequency is split into many orthogonal channels [39]. The nearby nodes simultaneously transmit their packets. This is used to reduce collisions and also increases the network capacity and throughput. These protocols are based on the channel assignment and also for reusing the control channel [39]. Each node is assigned to a specific channel and only uses that channel to connect with other nodes [39,40]. The benefit of this method is that it allows the reduction of the interference and of collisions occurring during data

transmission. The unnecessary channel switching is an issue. These protocols are subdivided into two branches. The details are given below.

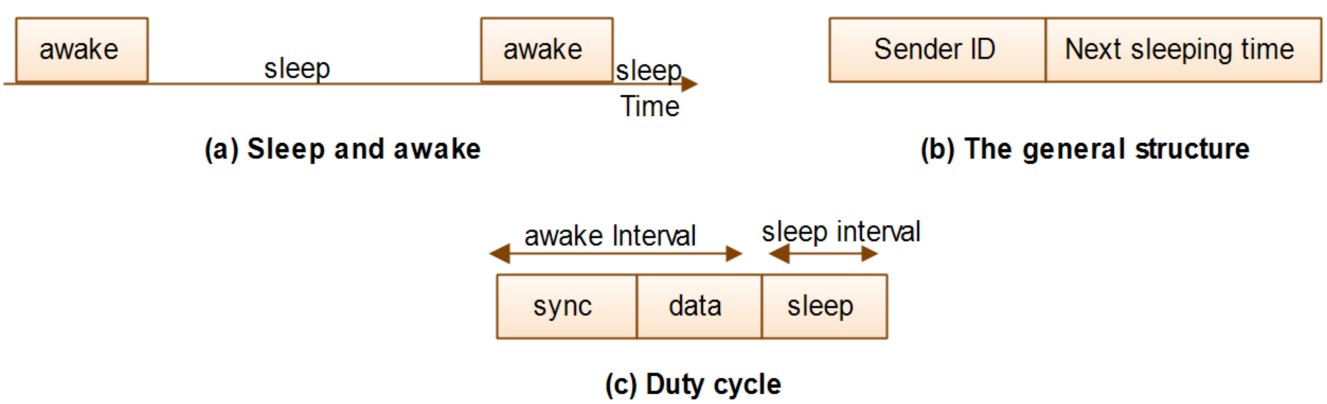

(a) Sleep and awake

(b) The general structure

(c) Duty cycle

**Figure 4.** The S-MAC protocol.

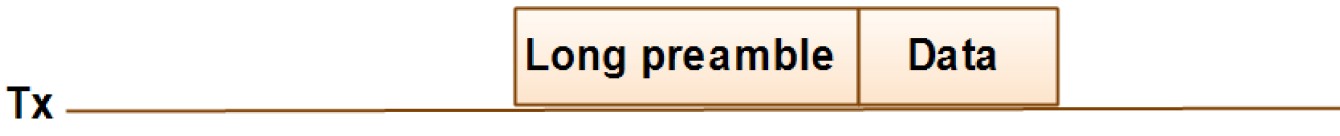

(a)Transmitter

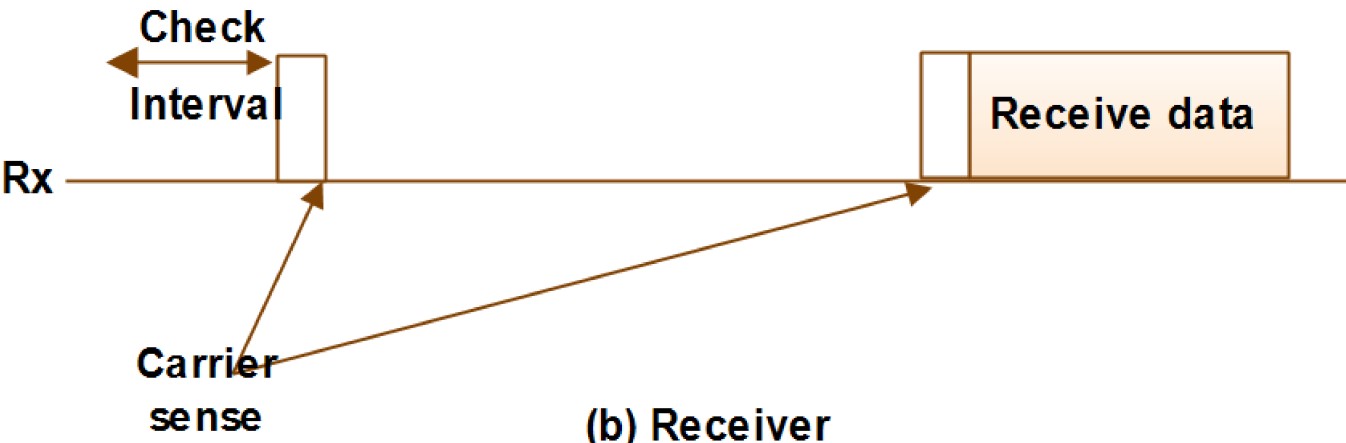

(b) Receiver

**Figure 5.** The B-MAC protocol.

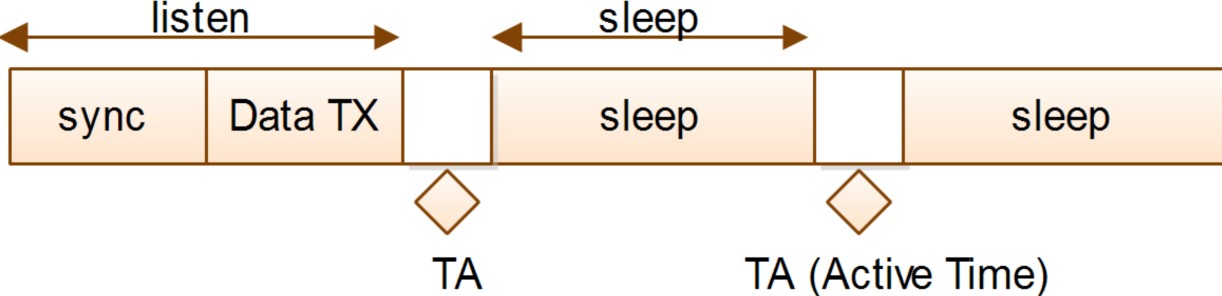

**Figure 6.** The sleep-wake up schedule of T-MAC protocol.

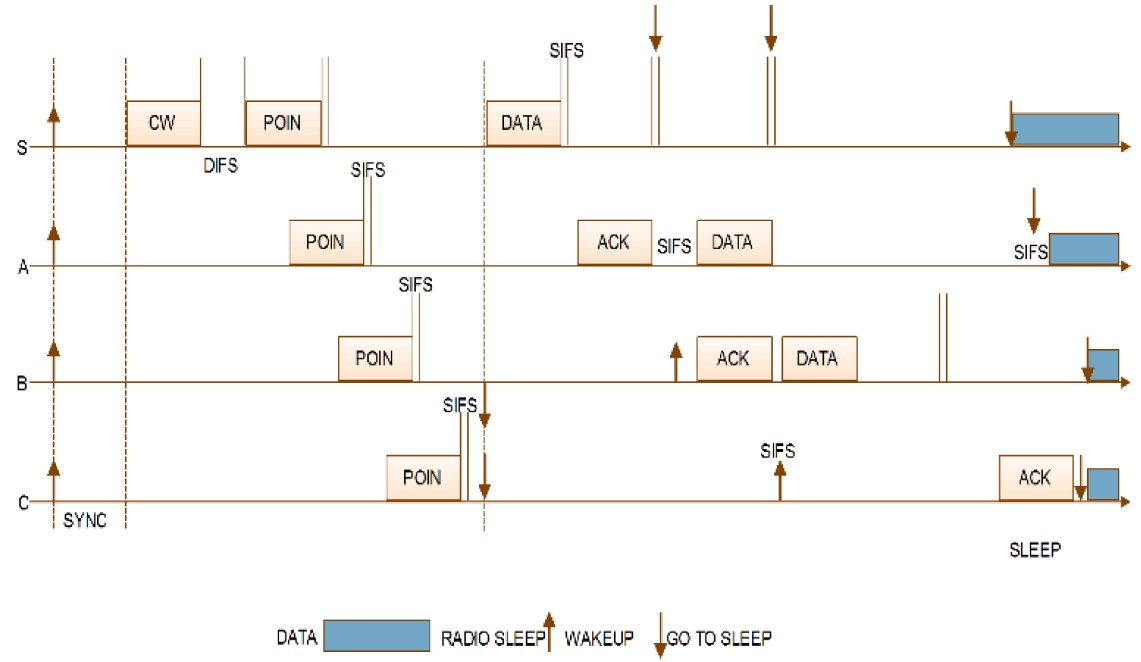

**Figure 7.** R-MAC overview.

Dynamic Multi-Channels Protocols

Y-MAC [41], is a scheduling-based approach used for WSNs. In this protocol, the time slots are assigned to the receivers instead of the senders. In each time slot, one data packet is transmitted. The frame architecture of the Y-MAC is shown in Figure 8. There is a longer end-to-end delay, due to the time and energy required by a radio chip for channel switching [41]. The channel conditions are not taken into the account by Y-MAC. The Multi-channel Medium Access Control (MuChMAC) protocol was introduced by Booms in [42]. A node chooses a channel for catching the upcoming timeslot as shown in Figure 9a [42]. A pseudo-random number generator uses the current time slot number and node ID to choose the next channel. MuChMAC transmits a preamble before each transmission, as depicted in Figure 9b [42]. A sender wakes up at the predicted time and sends several short preambles. This scheme uses a dynamic channel. This scheme does not offer any way to prevent switching to channels with poor conditions. The goal of this protocol is to achieve both high performance and energy efficiency under diverse traffic conditions. It is usually used in traffic monitoring applications. In [43], Efficient Multichannel MAC (EM-MAC) has been suggested to overcome the issues in the previous protocol. Since the channel numbers and the wake-up schedules are not expressly exchanged, the EM-MAC does not use a common control channel. Each node in the EM-MAC uses a common pseudorandom number generator to provide a channel number and a time for the subsequent wake-up event. The energy-efficient multichannel MAC is used for high-traffic applications [44].

The LMAC with multiple channels (MC-LMAC) [45] uses a similar approach. LMAC allots the timeslots to nodes for medium access. Channels and the timeslots are both assigned in MC-LMAC. Every node keeps track of its neighboring nodes' assignments. and data. Each column is a timeslot, and each row represents a channel. The channels are indicated by the gray cells. If a new node enters the system, it receives information about assignments from nearby nodes. Then, it chooses a cell if it is empty [44]. This protocol is also useful in traffic applications.

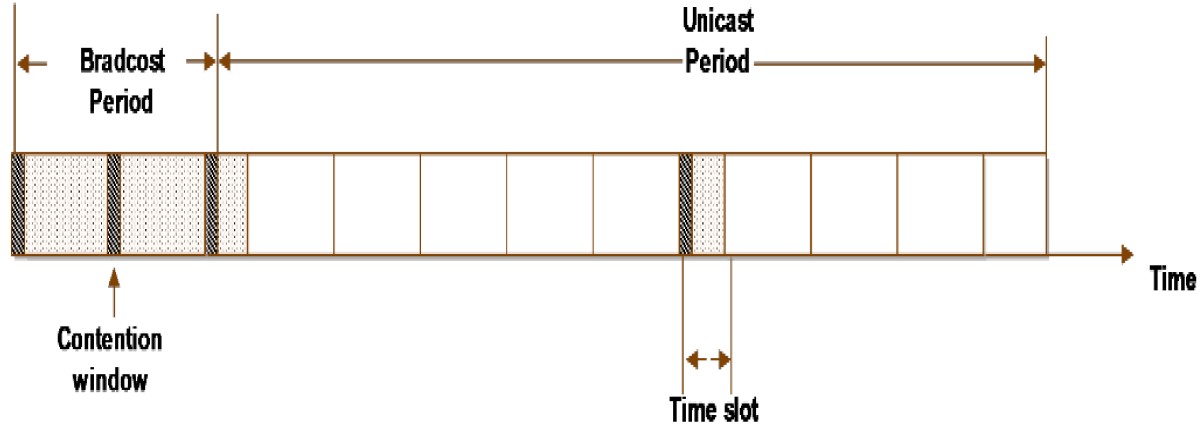

**Figure 8.** Y-MAC frame structure.

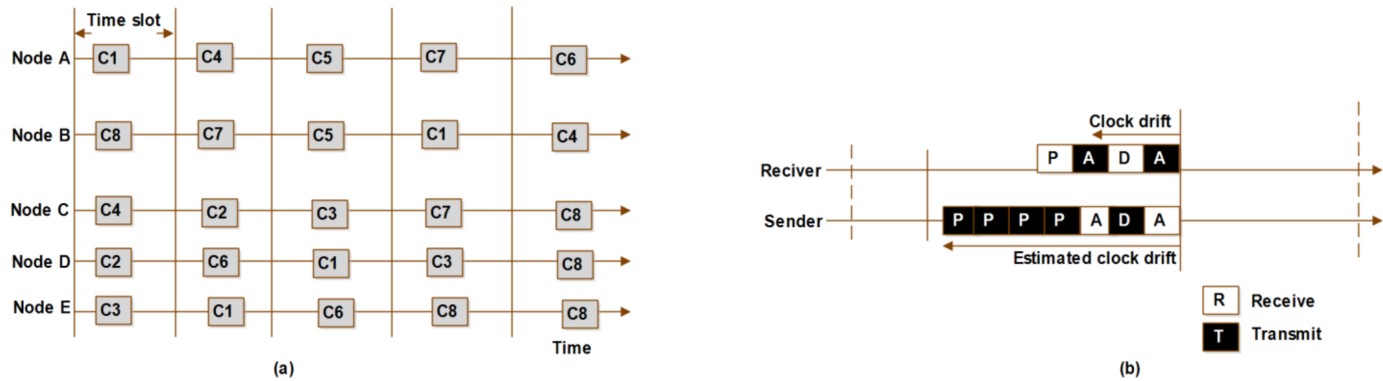

**Figure 9.** (**a**) Channel selection. (**b**) Preamble based communication of a MuChMAC.

Static Multi-Channel Protocols

Le et al. in [46], proposed a protocol based on control theory. This protocol uses clustering. In clustering, every node in a cluster uses the same channel for communication [44]. A network's initial channel is used for communication between all nodes. When the primary channel becomes overloaded, they progressively switch to the other channels [47]. Every node periodically broadcasts the data to check the channel load and if it is active. Each node calculates the likelihood that its nearby nodes will successfully acquire the channel. If the likelihood falls below a then certain threshold, the node changes its radio channel [47]. The channel switch is controlled by using this scheme. This protocol is was tested using MicaZ motes [46]. Wu et al. suggested a tree-based multi-channel protocol named (TMCP) [48]. A network was divided into several sub-trees with the least amount of intratree interference [48]. To do this, they assigned various channels to the nodes. These nodes are located and based on various branches. In this channel assignment, tree construction is performed. It is a challenging task to make effective broadcasts, since all nodes use the same channel for communication. This protocol is used for data collection applications. Gupta et al. in [49], proposed a clustering strategy. Each cluster is given a unique channel

for communication. There is no need for channel organization because the logical assignment of channels and the creation of clusters are based on relatively basic rules [48]. It is a challenging task to adapt this system for handling various applications. The comparison of multi-channel protocols for the IoT is shown in Table 3. The channel assignment is difficult as compared to dynamic approaches since the fixed channels are assigned to the nodes. It is more effective than fixed channel assignment approaches because nodes can still switch their interfaces to other channels to communicate with the nodes on these channels. Each of the protocols is discussed along with implementation and the key application areas [48].

**Table 3.** The comparison of various channel assignment methods.

| Protocol | Assignment | Control Channel | Implementation | Synchronization | Medium Access | Channel Model | Objective | Applications |
|----------|-----------|-----------------|----------------|-----------------|---------------|---------------|-----------|--------------|
| Multi Channael clustering [49] | Fixed | No | Distributed | No | TDMA/ CSMA | Orthogonal | Improve the energy efficiency | Forest fire detection |
| TMCP [48] | Fixed | No | Distributed | No | - | Orthogonal | Efficient data collection | Data collection applications |
| MuChMAC [42] | Dynamic | No | Distributed | Required | XMAC | Orthogonal | Improve bandwidth | Traffic monitoring |
| MC-LMAC [44] | Dynamic | No | Distributed | Required | Slotted | Overlapping | Fast converge-cast | Traffic applications |
| EM-MAC [43] | Dynamic | No | Distributed | Required | Slotted | Orthogonal | Improve energy efficiency | Traffic monitoring |

### 3.3. Contention-Free Protocols

In contention-free protocols, the messages do not collide during execution. Due to communication, the time slots are allocated without contention. Figure 10 shows the flow chart for contention-free protocols. These protocols are further divided into different classes. The details are given below.

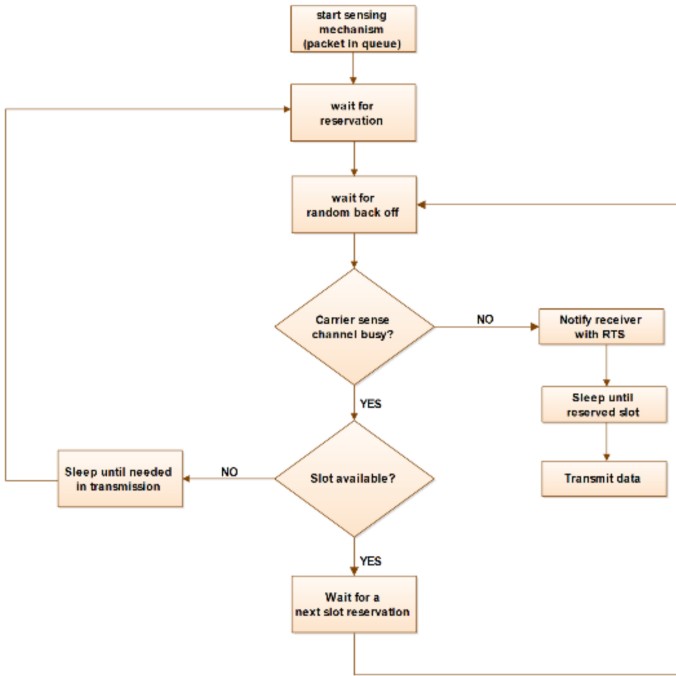

**Figure 10.** The structure of contention-free protocols.

### 3.4. Schedule-Based MAC Protocols

A central point grants access to a shared medium, and broadcasts use a schedule. The nodes receive deterministic access to the media and can offer delay-bounded services in line with the scheduling [49]. TDMA is a more energy-efficient schedule-based MAC

protocol because it is intrinsically collision-free and does not engage in pointless idle listening. The contention-free protocols are frequently built on the wireless medium. The TDMA presumes that all of the sensor nodes are time-synchronized [49]. However, this approach typically has bad experiences due to transmission latency and changes in network topology. These protocols assign the collision-free linkages to adjacent nodes during the initialization phase. However, based on the spread spectrum, the links are assigned as FDM [50]. The time is divided into slots. All of the nearby nodes are then assigned time slots. The participant's permission to access the resources at set times, however, is controlled by a schedule [50]. These are further subdivided into the following categories.

### 3.4.1. Centralized Protocols

The nodes are centrally programmed according to the time slots [50]. The base station (BS) is responsible for the scheduling. The cluster head assigns the scheduling periods for each node if any node joins or departs the network. For instance; Traffic-Adaptive Medium Access (TRAMA) is introduced in [51]. In this protocol, energy-efficient collision-free channel access enables the nodes to build an on-demand schedule to access a single channel. It is assumed that all nodes are in sync concerning time. There are various cycles within time, and each cycle consists of a random access period. In TRAMA, the Neighbor protocol (NP), Scheduled Exchange Protocol (SEP), and Adaptive Election Algorithm (AEA) are the key elements. The key application for this protocol is traffic monitoring applications. A lightweight Medium Access Control Protocol (LMAC) in [52] is proposed. It is used to reduce the number of transceivers. Switches and make the sleep interval. [52]. We have compared various schedule-based MAC protocols along with performance indicators, such as traffic adoption, time latency, overhead, and complexity in Table 4. In this table, we examined that the latency and the complexity of few protocols are high. Few protocols are good when there is a change in topology. They will automatically adopted it.

**Table 4.** Schedule based protocols comparison with indicators.

| Protocol | Traffic Adaptive | Adaptivity to Changes | Latency | Overhead | Complexity |
|---|---|---|---|---|---|
| TRAMA [51] | Yes | Good | High | Neighbor protocol and schedule transmission | High |
| LMAC [52] | Yes | Moderate | High | Network setup and control message | Low |
| EMAC [53] | No | Good | High | Timeslot selection and CR, TC of each time slot | Low |

### 3.4.2. Distributed Protocols

Gang Lu et al. in [27] discussed a distributed protocol, which adjusted itself for changes in bandwidth or topology. The DMAC is a low-latency and high-energy-efficient model specially developed for the WSN. The data forwarding process stops at a node whose next step toward the sink is outside of the overhearing range because of the limited coverage. In DMAC, the sending, receiving, and sleep periods are separated into various intervals. The sending/receiving times are equal in duration. A node advances its wake-up schedule from the sink schedule in the data-gathering tree by the depth 'd' from it to the sink node. To save energy and reduce latency, a node enables sending right after receiving packets. A node raises its and other duty cycles on the multi-hop path when it has several packets to send [27]. The DMAC makes better use of (MTS) packets and sends a request MTS packet to its parent node in the data gathering tree to wake up earlier than usual [27]. It lacks end-to-end data transfer dependability, making it unusable for real-time applications without improvement. The Distributed Energy-aware MAC (DE-MAC) protocol is in advanced form and used for the WSN [54]. It is a TDMA-based protocol that is used to extend the network lifetime by treating low-power nodes differently in a distributed manner [55]. Each node is initially given two timeslots for transmission [56]. If the two nodes

do not broadcast in the same slot, the packet will be lost due to collision [56]. All nodes are required to listen to every packet delivered by every other node. This causes an overhearing problem. The DNIB algorithm in [57], is composed of three phases, for instance; slot assignment, update, and recovery.

### 3.5. Hybrid Protocols

The hybrid protocol is a combination of contention and scheduling protocols. Low-power and low-rate wireless networks can be implemented using hybrid CSMA/CA and TDMA protocols [58]. The schedule-based MAC protocol is known as TDMA. It is the best protocol for preventing collision issues during high traffic [58]. The ADV-MAC in is helpful to reduce the energy and is used for idle listening while maintaining the throughput and latency [59]. It offers synchronization during transmission [60], [61] Banerjee et al. reported in [62] that the main cause of the standard's low data rate and dependability is the unneeded packet drops, which occur during data transmission as a result of beacon super-frame broadcasting. There is not enough time available for data transmission throughout the super-frame. They incorporate a back-off freezing mechanism, which causes the back-off counter to freeze at any time. There is not enough time for the data transmission within that super-frame length [62]. Z-MAC in [63] is a CSMA/TDMA protocol. It easily adjusts to the degree of network contention. The Z-MAC uses CSMA for two-hop neighbors. It is used to avoid the hidden terminal problem [64]. The Distributed RAND is used to build the channel reuse [65,66]. This scheduling method assigns slots to each node in the network. Many nodes can own the same slot, but only a particular node can transmit [67–69]. The non-owners receive priority in turn [70]. The NoPSM in [71] is a hybrid MAC protocol developed for WSN. NoPSM functions as a block of data in packets. A data packet makes up the data block. The senders keep track of the beginning and end of each block transmission. A stream of simultaneous transmission in a WSN2. Typically, a block of data is waiting to be transmitted. As soon as node S0 observes the current traffic flow [71], it expands the source node's identifying information (source ID) from the current flows [71]. In [72], Nurzaman et al. examined large-scale networks for the IoT by contrasting the effectiveness of MAC protocols that are reservation-based, contention-based, and hybrid-based. Table 5 presents a comparison of hybrid protocols along with energy efficiency, throughput, latency, and fairness. In this table, we examine which protocol is most suitable for the traffic application and has low latency.

**Table 5.** Comparison of hybrid protocols.

| Protocol | Energy Efficiency | Throughput | Latency | Fairness | Application |
|---|---|---|---|---|---|
| ADV-MAC [39] | Best | Best | Low | Best | Traffic |
| BSMAC [62] | Best | Best | Low | Good | Traffic |
| Z-MAC [64] | Best | Best | Low | Best | Traffic |

## 4. The Duty Cycle Protocols

Figure 11 demonstrates the proposed classification for the duty cycle protocols. These protocols are classified into a synchronous low-duty cycle and asynchronous low-duty cycle protocols. The asynchronous low-duty cycle protocols are further subdivided into sender-initiated low-power listening and receiver-initiated low-power listening. Duty cycling has been utilized for a long time in many different types of equipment to conserve energy [38]. Duty cycling is a key requirement for building applications in IoT. It works even if nodes operate for longer than a few days before recharging the battery [38].

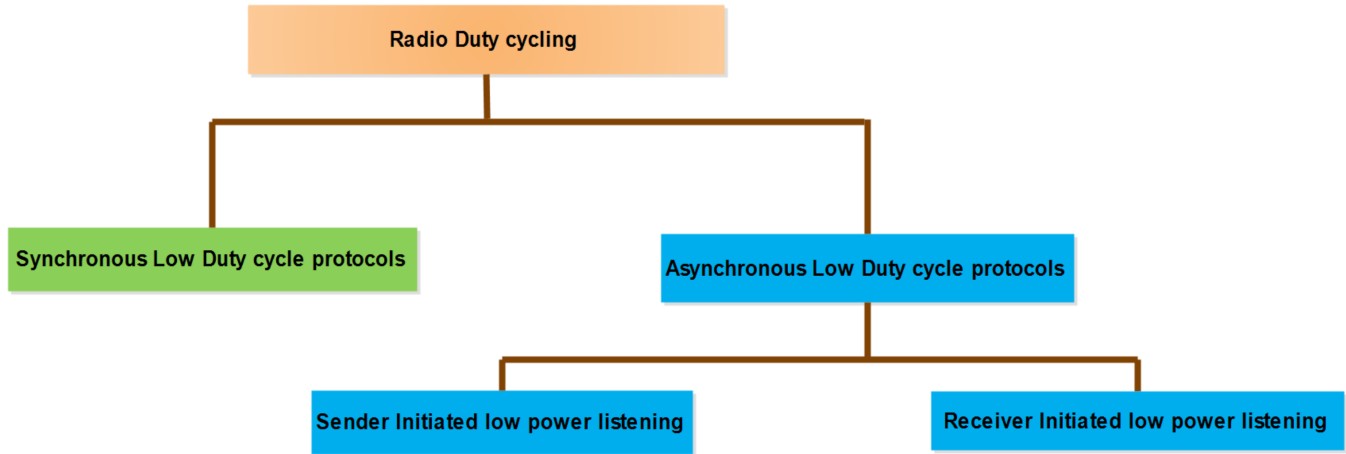

**Figure 11.** Proposed classification for radio duty cycle protocols.

### 4.1. Synchronous Low Duty Cycle Protocols

The nodes maintain common time references during coordination [38]. The primary goal of SCP-MAC in [29] is to reduce energy consumption through the use of channel polling and scheduling. For instance, the SCP-MAC uses low-power listening (LPL). The LPL in SCP-MAC synchronizes the polling times of all the nodes that are close to it. The key steps are as follows. Any sensor node that initially wants to join the network must first determine when it is scheduled to wake up. It has to use a low-power listening protocol. It transmits a preamble message that is lengthy enough to synchronize with the network's sensor node's wake-up schedule while using a low-power listening duty cycle. The Adaptive channel polling improves the SCP-MAC by introducing slots to boost the overall throughput. In DW-MAC [25], the sensor nodes awaken from their sleep mode and transmit a packet when there is a need to transmit the data. When the traffic increases, this demand particularly raises the channel capacity. The DW-MAC protocol is used to achieve minimal delivery delay even under traffic loads. It contains unicast and broadcast traffic. By utilizing the SCH carriers, the DW-MAC protocol creates a one-to-one mapping between two nodes. The RTS, an SCH frame contains the destination node address; hence, this SCH wakes just the intended recipient to whom a sensor node must pass the data, thereby reducing energy usage by avoiding unwanted wake-ups. The P-MAC in [65] proposes to deliver multiple messages every duty cycle, building on R-MAC [26]. P-MAC uses grade division and scheduling assignments to partition the network around the sink node (GDSA). Each node establishes its schedule by the grade to which it belongs. Nodes in the same grade will continue to have the same scheduling time. The lower and upper grades in this program are spaced out. To decrease the network latency, the P-MAC uses pipelining to forward packets from higher to lower grades. Table 6 demonstrates the comparison of synchronous duty cycle protocols. In this table, we examine each protocol algorithm's key design issues, strengths, and weaknesses, and the applications are stated separately. For instance, the P-Mac is not useful in reducing collisions and the SCP-MAC helps to reduce them. The AD-MAC, BSMAC, and Z-MAC seem to have good throughput.

### 4.2. Asynchronous Low-Duty Cycle Protocols

LPL (Low-Power Listening) protocols can also be divided into sender-initiated and receiver-initiated categories. The explanation details are given below.

**Table 6.** Key The synchronous duty cycle protocols comparison.

| Protocol | P-MAC [65] | SCP-MAC [29] | DW-MAC [25] |
|---|---|---|---|
| Mechanism | Scheduling | Adoptive duty cycle | Scheduling |
| Strategy | Listening/sleeping | Channel polling | Demand wake duty cycle |
| Collision | Unable to reduce the collisions | Reduced | One to one mapping |
| Applications | • Continuous monitoring<br>• Multi-hop communication | • Environment monitoring applications<br>• Multi-hop communication | • Environment monitoring |
| Key design | • Support pipeline | • It combines the strengths of channel polling and scheduling | • Control/ scheduling |
| Strengths | • Cross-layer | • Less scheduling maintenance | Low delay |
| Weakness | • Does not exploit linear topology in the network | • Listen interval long | Long idle listening |

### 4.2.1. Sender-Initiated Low-Power Listening Protocols

Preamble sampling is the fundamental method used for the sender-initiated low-power listening protocols. The sender transmits a preamble to signal for communication. The receiver awakens in active mode to receive a preamble signal. When the preamble broadcast ceases after being recognized, the receiver responds positively to the sender. Good examples are; Wise-MAC [6], B-MAC [7], and X-MAC [8].

### 4.2.2. Receiver-Initiated Low-Power Listening Protocols

Each node awakens to check for new data. Each wake-up event is followed by the emission of a beacon in the network. This beacon alerts the nearby residents that it is prepared to receive the incoming data. The receiver keeps listening to a channel for a short while after the beacon has been transmitted. When a node enters the active state and has data ready to send, it silently watches for a beacon from the target receiver. The transmitter immediately begins delivering the data after the beacon is caught and waits until the frame arrives. An acknowledgment confirms the reception of the data. The receiver nodes go into sleep mode if the sender does not send any data after sending the beacon. The sender and receiver then start their cycles [66]. The acknowledgment beacon is used by the sender as a Ready-To-Receive (RTR) indicator after the data transmission and if there are still more data packets to send. After sending a beacon, the receiver goes into sleep mode if there is no incoming data from the sender. The result is that the typical MAC's duty cycle becomes static. The main change made by A-MAC [67] is to being receiver-initiated. A-MAC in particular uses a brief packet called HACK to acknowledge the beacon. This acknowledgment's main function is to rapidly alert the recipient to any pending communications. The receiver immediately goes to sleep if the beacon does not cause a HACK packet to be sent. As a result, after each unanswered beacon, the receiver uses less energy in idle listening. The receiver is still able to recognize that there is waiting traffic even if various HACK packets from various senders conflict and keeps the radio on. The LPP in [68] is used for asynchronous network awakening from a deep slumber and is also included in A-MAC. If there is no activity on the network, the nodes may go into a deep sleep and only occasionally wake up to send out beacons [69]. A node switches on and keeps its radio active, listening for beacons, whenever an event that should wake up the network occurs [70–72]. Wake-up requests are sent in response to these beacons [73]. Such requests will be propagated by nodes that accept them, gradually waking up the entire network [74]. The maximum amount of time needed for an asynchronous network to wake up depends

on the deep sleep nodes' beacon frequency [75]. A synchronous duty cycle protocol with receiver-initiated data transfer is called SRI-MAC [76,77]

## 5. Key Challenges & the Future Research Directions

In this section, we discuss several challenges for the IoT MAC and RDC protocols from these aspects.

### 5.1. Scalability and Large Network Size

Scalability is a key characteristic of any organization, model, or system [78–80]. The Adhoc and wireless sensor networks depend upon primarily multi-hop and peer-to-peer communication without centralized control. The MAC protocols should not impose undesirable computation weight on a system. It could possess extensive memory for the persevering the state information. To improve the scalability, a hybrid scalable MAC protocol for IoT networks is available which can accommodate a large number of IoT devices without much computational overhead [81]. However, due to the unpredicted topology changes and dynamic characteristics of IoT, the MAC protocols targeted for IoT networks are not suitable. The development of efficient clustering of IoT devices can help to reduce the number of transmissions from various IoT devices. In addition, by considering the mobility and limited communication time with IoT devices, the prioritized access to the channel enhances the fairness and the scalability of the overall system [82].

### 5.2. Energy Consumption and Network Lifetime

The IoT devices are battery-operated [83]. Therefore, they have power for a limited time [84]. For a longer network lifetime, new and efficient energy consumption methods are be required. Although in the literature, many studies have been performed on the development of efficient energy consumption, there are several challenges to overcome energy consumption, such as idle listening, overhearing, and collisions. Solar power is another option for energy harvesting using natural sources. However, on rainy days or in winter, solar power is normally unavailable. Moreover, when a collision occurs, the packets need to be retransmitted. A large amount of energy is wasted due to this problem. Possible solutions would be to develop dynamic sleep and wakeup-based duty cycling. These advanced mechanisms could help to reduce unnecessary energy wastage in the monitoring of applications. However, reducing the number of transmissions, solving the hidden and exposed nodes problem and the proper handling of collisions can help to reduce energy consumption and also enhance the network's lifetime.

### 5.3. Interpretability

Interpretability is the ability to connect different systems and applications. The interpretability includes data transmission, data access, and also cross-organizational collaboration regardless of data origin. In the IoT, interpretability is a major concern because of the large number of different platforms used in IoT. How to connect and communicate with various platforms is the key concern. With the current evolution, the devices need several connectivity technologies. The interpretability between the devices should be perfect so that they can achieve their desired goals. Most of the current MAC protocols are designed for the specific transceiver hardware and usually assume that all nodes (sensor nodes and subscriber stations) in the network are homogeneous. As the devices are diverse, therefore, new MAC protocols need to be developed for these scenarios where nodes have nonhomogenous capabilities and constraints.

### 5.4. Fairness

Fairness participates virtually in node resource sharing. The crucial components are bandwidth distribution, channel assignment, and power control. The distribution of resources, whether in terms of bandwidth, power, throughput energy, or quality of service

(QoS), must be equitable. The QoS could be affected by unequal resource distribution. Energy waste and fairness issues should be addressed and solved in the future.

## 6. Conclusions

In this paper, we presented a comprehensive review of a specific class of protocols for IoT. The IoT MAC and RDC protocols are effective methods used to overcome the energy efficiency challenges in IoT. We proposed a unique classification and classified the most recent protocols into three categories: contention-based, contention-free, and hybrid-based MAC protocols. We classified the RDC protocols separately. In this catalog, we have surveyed the most recent research from both academia and industry and numerous protocols are outlined. In this review, we have discussed the strengths, weaknesses, and application of each protocol. The ultimate goal is to highlight the significance of recent research efforts in this direction. Finally, we have summarized various open research issues and the main challenges of the IoT OS, such as scalability, energy management, interpretability and fairness.

**Author Contributions:** Conceptualization, F.A.; Methodology, F.A.; Formal Analysis, R.A., Investigation, M.A.A.; Resources, S.K.; Writing—Original Draft Preparation, F.A.; Supervision, M.A.A.; writing—review and editing, F.A., S.K. and R.A. All authors have read and agreed to the published version of the manuscript.

**Funding:** National Natural Science Foundation of China (NSFC) (No. U21A20146). Open Foundation Project of State Key Laboratory of Power System and Generation Equipment (No. SKLD21KM09). Synergy Innovation Program of Anhui Polytechnic University and Jiujiang District (No.2021cyxta2). Wuhu Science and Technology Project (No.2021cg19).

**Institutional Review Board Statement:** Not applicable.

**Informed Consent Statement:** Not applicable.

**Data Availability Statement:** Not applicable.

**Acknowledgments:** We thank our families and colleagues who provided us with moral support.

**Conflicts of Interest:** The authors declare no conflict of interest.

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
