# Peer review of "An Overview of Medium Access Control and Radio Duty Cycling Protocols for Internet of Things"

_electronics, doi:10.3390/electronics11233873_

Round 1

Reviewer 1 Report

Dear authors,

You've collected and curated a good amount of bibliographic references for MAC and Radio duty cycling protocols. 

The general structure of the paper is okay. However, the narrative is very complicated and monolithic. Also, there are major English faults, and the referencing style is incorrect. Please improve the figure's resolution, font size, and description. 

I'd recommend following the guidance of well-known bibliography on technicalities and the form of writing papers, perhaps an editor would help in giving the desired shape. 

Regards,

Anonymous Reviewer. 

Reviewer 2 Report

The topic discussed is attractive and very current. However, to improve the manuscript, I think it is important for authors to consider the following:

1)Correct some grammar and typing errors.

2)Remove Table 3 because it is not important for the manuscript. Perhaps the authors could consider including an appendix with a list of acronyms.

3)The various protocols seem to be discussed, but I do not feel that they are well contextualized. It would be interesting to include a section where they are also discussed under different scenarios. For example, smart grid, smart cities, e-healthcare, are initially discussed, but it is not highlighted, which protocol is better suited to meet the requirements of one scenario rather than another.

Reviewer 3 Report

·      The paper describes that in the IoT physical objects are connected to the Internet through sensors and actuators with smart cities, smart healthcare and smart highways ae all examples of the IoT with conceptual diagram as shown in Figure 1.

·      The connectivity is achieved through cloud server using different communication technologies such as WiFi IEEE 802.11, Bluetooth IEEE802.15.1, IEEE 802.15.4/802.15.4e ZigBee, and IEEE 802.11p WVE sub 1GHz, etc.

·      The Medium Access Control protocol ensures successful transmission of data which is characterized as having high throughput, network scalability, capable of reducing latency, and energy constrained.

·      The MAC uses Radio Duty Cycling protocols for efficiently managing the energy of devices  and so MAC RDC is investigated for future applications of IoT applications.

·      The paper is having the following contributions:

§  The RDC perspective for future IoT.

§  Finally, the challenges, open issues, and recommendations for future work are pro-103 vided at the end of this article, which may help to improve available schemes or de-104 signing of more innovative MAC/RDC protocols for IoT.

·      Claiming TinyOS and OpenWSN as the most common open source, saying that these ‘These ora are commonly used in more sophisticated, industrial, and healthcare IoT applications’. Very commonly used terms need be avoided unless they are technology representative

·      ‘To the best of our knowledge and 86 as depicted in Table II also, other surveys have not considered all MAC and RDC protocols 87 using OpenWSN or TinyOS’, the authors claim, saying that recently a lot of researchers have focused on the design and implementation of MAC and RDC protocols for WSN.

·      The classification catalog given in Fig. 2 shows that MAC protocols - contention-based (Fig. 3), contention-free (Fig. 13), and hybrid protocols ) and RDC protocols for IoT

·      B. Contention-Free protocols onwards we see B. Hybrid Protocols, B. Duty Cycle protocols, II) Asynchronous low-duty cycle protocols, a kind of misleading flow is visible

·      4. Key Challenges & Future Research Directions

4.1 Scalability has not been discussed, rather is like a description

 Query 1.  It is not presented in literature-based style, could have been as key challenges”

4.2. Energy Management

Here as part of this section some key challenges should have been discussed. For example, the sentence “The protocols used nowadays are solely concerned with throughput or 590 energy efficiency” does not provide discussion, rather it is about information report. The mobile nodes can obtain energy from their environment by using energy harvesting. ‘For software-defined IoT solutions, highly energy-efficient techniques must be developed. Scalable and context-aware data and services are provided by these solutions’ shows it very descriptive.

 Query 2.  No results and no critical analysis is presented, Overview is about synthesizing and combining relevant information data as a result of existing reviews to present solutions that are better and readers-impressive

4.2. Mobility

The mobility is covered very descriptively, saying that the ‘The parameters of the MAC layer could be dramatically altered by this mobility feature. It also significantly affects the choice of relays and the distribution of electricity among users. The main issues that make relay selection more difficult include channel conditions, changes in network architecture, and relay selection in dynamic networks [62]. The research community should therefore regard it as a design concept’ shows very descriptive and not well-connected way of presentation by repeating 4.2 similar to B- earlier.

Is the subject matter presented in a comprehensive manner?

·      The 25-page paper is presented with not an enough level of flow to genuinely support the title An Overview of Medium Access Control and Radio Duty Cycling Protocols for Internet of Things”.

·      There should some theory support and related explanation covered comprehensively, justifying the contribution to the body of knowledge in terms of implementing them for Medium Access Control and Radio Duty Cycling Protocols for Internet of Things.

·      The 25-page paper seems to be more of a report than a paper, and the conclusion is equally inclusive to justify publication of an overview as a synthesis of information data from reviews. of

Are the references provided applicable and sufficient?

·      The authors take support from seventy eight (78) mostly recent articles from diverse journals references with none from MDPI algorithms.

·      The whole presentation does not justify a well-deserved support to the title of An Overview of Medium Access Control and Radio Duty Cycling Protocols for Internet of Things”.

Round 2

Reviewer 2 Report

It seems that the authors did what I requested, improving the paper by editing some sections and tables to contextualize the described protocols. Therefore, the paper is acceptable.

Reviewer 3 Report

I have verified that the authors have responded, and I am satisfied with the authors’ response.